# The Influence of the Presence of Borax and NaCl on Water Absorption Pattern during Sturgeon Caviar (*Acipenser transmontanus*) Storage

**DOI:** 10.3390/s20247174

**Published:** 2020-12-15

**Authors:** Massimo Brambilla, Marina Buccheri, Maurizio Grassi, Annamaria Stellari, Mario Pazzaglia, Elio Romano, Tiziana M. P. Cattaneo

**Affiliations:** 1Consiglio per la Ricerca in Agricoltura e L’analisi Dell’economia Agraria–CREA–Centro di Ricerca Ingegneria e Trasformazioni Agroalimentari, Via Milano, 43–24047 Treviglio (Bergamo), Italy; massimo.brambilla@crea.gov.it (M.B.); annamaria.stellari@crea.gov.it (A.S.); elio.romano@crea.gov.it (E.R.); 2Consiglio per la Ricerca in Agricoltura e L’analisi Dell’economia Agraria–CREA–Centro di Ricerca Ingegneria e Trasformazioni Agroalimentari, Via G. Venezian, 26–20133 Milano, Italy; marina.buccheri@crea.gov.it (M.B.); maurizio.grassi@crea.gov.it (M.G.); 3Agroittica Lombarda S.p.A., Via Kennedy, 25012 Calvisano Brescia, Italy; mario.pazzaglia@agroittica.it

**Keywords:** NIR spectroscopy, aquaphotomics, caviar quality, portable instrumentation, sustainability, qualitative determination, food chain monitoring

## Abstract

Sturgeon caviar quality relies not only on the perfect dosage of the ingredients but also on the long sturgeon breeding cycle (about 12–15 years) and the exact timing of the egg extraction. For the improvement and the promotion of Italian caviar, the development of an analytical system dedicated to fish products, and caviar, in particular, is fundamental. The use of near-infrared spectrometry (NIRS) technology is auspicious. The aquaphotomics approach proved to be an adequate analytical tool to highlight, in real-time, the differences in caviar quality stored with, or without, borax as a preservative. Seventy-five sturgeon caviar (*Acipenser transmontanus*) samples underwent spectral NIR characterization using a microNIR1700 in the 900–1700 nm range. Data processing was carried out according to the literature. Tenderometric and sensory analyses were also carried out in parallel. The results suggest that a process line under strict control and monitoring can result in high-quality caviar without any other preservative than salt. The challenge of producing caviar without any potentially-toxic preservatives could now be a reality. NIR spectroscopy and aquaphotomics can be, in the future, non-invasive methods to monitor the whole production chain.

## 1. Introduction

Sturgeon caviar (from now on, called caviar) is a food product whose preparation appears extremely simple, requiring sturgeon eggs and salt. However, its quality relies not only on the perfect dosage of the ingredients, but also on: (i) the long sturgeon breeding cycle (about 12–15 years), and (ii) the exact timing of the egg extraction. Various techniques establish how to process sturgeon roe to caviar, as [1] reported. These processing techniques significantly impact product composition and quality and, thereby, marketability [2,3,4,5]. The preparation processes differ mainly on the salt content (NaCl), ranging from 3.2% to almost 11.8% among the producers. The high salt content results in dehydration, which increases the concentration of lipids and proteins in a linear pattern. 

Nowadays, sturgeon breeding in Lombardy (Italy) is mainly oriented to caviar production and has attained global manufacturing leadership in this field. The Lombard caviar production exceeds 40 tons/year, 90% of which represents the export for a value of about 15,000,000 euros. The appearance on the international market of products from countries with high production capacities and lower costs could represent a new obstacle for Lombard businesses and livestock. Not being able to compete, in terms of price, it is necessary to continue investing in product quality and innovation. Given the high costs of the raw materials, the caviar industry must offer an end product with the following features: (i) stable quality; (ii) sensory attributes well characterized; and (iii) adequate documental support from reliable and low-cost methods of investigation.

Salting is considered the more delicate phase in light of the high variability of the roe. Compared to the wild one, the farming environment reduces the eggs’ variability; however, the inclusion in the regional context of new species of sturgeon and new farming conditions makes it difficult to calibrate the correct salting mixture for each circumstance. Italian business experiences have shown that different production batches, subjected to the same dosage, could result in different salt contents of the final product. During the product’s maturation, variable loss of brine occurs at varying of the salt mixture.

For the improvement and the promotion of Italian caviar, the development of an analytical system dedicated to fish products and caviar is fundamental, and the use of near-infrared spectrometry (NIRS) technology is auspicious, not only for the quality of the final product, but also as an innovative control tool along the production line [6,7]. The European Union and the national legislations still allow some boron compounds (i.e., E284: boric acid-E285: sodium tetraborate) as preserving agents in caviar; however, recently, the Codex Alimentarius and some national regulations have pushed towards a borax-free product. Although the EFSA (European Food Safety Authority), in its latest scientific opinion of 2013 [8], confirmed the limited use of borax for caviar preparation, the international market prefers a borax-free product. Many countries have already prohibited the use of borax. The realization of a product free of preservatives improves its safety and its end quality.

Among the different NIR technologies and procedures, several papers have reported the use of aquaphotomics as an immediate indicator of biological systems changes through the study of water pattern modification in the 1300–1500 nm NIR region [9]. This work aims to study the effect of borax on caviar storage, to develop a borax-free product with suitable organoleptic characteristics throughout the shelf life. The aquaphotomics approach was chosen as an adequate analytical tool to identify borax as a preservative in caviar during the curing process. A rapid and objective method that allows the discrimination of the two treatments (salt+borax and borax-free) is a prerequisite to improve caviar’s quality and safety at the end of the production chain. Fast detection and measurement of the borax presence would be intrinsically part of an innovation process of total quality control. 

## 2. Materials and Methods

Materials: Spectra from seventy-five caviar (*Acipenser transmontanus*) samples were collected from Agroittica Lombarda S.p.A. (Calvisano, Brescia, IT), an Italian caviar factory known at the international level. Spectra were collected at scheduled times: before treatment (“no salt” samples) and after treatment, with 3.5% NaCl or a mixture of 3.5% NaCl + 0.04‰ sodium tetraborate at the time 0 and after 90, 150, and 210 days of storage. On the same samples, the tenderometric analysis was carried out using a Texture Analyzer TA 32 XT PLUS II (Stable Micro Systems Ltd., Godalming, Surrey GU7 1YL, UK), modified to test individual eggs at the constant temperature of 0 °C. The values of egg consistency and the distance and time the instrument took from the beginning of the measurement to the instant of the egg breakage were measured on every single egg (30 eggs/sample).

NIR measurements took place directly on caviar samples before the packaging in jars (50 g each, time 0), and afterwards, at the opening of the jars during storage. Visual inspection detected the potential presence of molds, and a panel of 20 expert panelists carried out the sensory analysis on the caviar samples at the end of the storage period at the factory headquarter. 

Spectroscopy and Chemometrics: A portable MicroNIR 1700 spectrometer (VIAVI Solutions Italia SRL, Monza, Italy) was used for the spectra collection in the range of 900 to 1700 nm (200 scans; 128 points; three replicates; reflectance mode, resolution: 6.0 nm; signal/noise ratio—S/N: 25,000). The spectral data, converted in absorbance, were preprocessed according to [10] to verify the suitability of the holistic aquaphotomics approach in highlighting the differences arising between borax and no borax samples during the storage. Excel spreadsheet (Office 365, Microsoft Corporation, Redmond, WA, USA) and MINITAB 17.0 statistical software (Minitab Inc., State College, PA, USA) were used for data processing. Repeatability between replicates was verified. Aquagrams were built up at time 0 and during storage (at 90, 150, and 210 days after the jar packaging), both for “borax” and “no borax” sets of samples. Principal component analysis (PCA) was applied (95% of confidence level) to the whole dataset, selecting the wavelength regions from 1300 to 1550 nm. Outlier detection in the multivariate space was carried out using the Mahalanobis distance criteria: observations falling above the critical distance were labeled as outliers and removed. The principal component analysis allowed the extraction of useful information from the dataset, the exploration of its structure, and the global correlation of the variables. Subsequently, using the most uncorrelated frequencies (i.e., 1342, 1374, and 1426 nm), the dataset underwent linear discriminant analysis (LDA) [11], one of the most used classification procedures, to assess the discrimination power of the aquaphotomics approach between borax and no borax samples. The dataset underwent splitting into two datasets: a calibration set, containing 2/3 of the observations chosen randomly, and a validation set, made up of the remaining 1/3.

## 3. Results and Discussion

Some examples of the quality of the collected spectra of sturgeon caviar samples are shown in Figure 1. 

The ranges of the variability of the main constituents of the analyzed samples are reported in Table 1. It is essential to consider that the composition can vary by individual constituent by about five percentage points, within the same species, except for the ash content. These data were already presented during the 6th Italian Symposium NIRItalia 2014 [12], as preliminary results.

The aquaphotomics approach was applied for the first time in the present work as a holistic approach to point out whether borax’s addition affects the caviar storage and the quality of the final product. Figure 2 reports the repeatability of measurements; the aquagrams of the samples before and after salt (NaCl) addition are shown (three replicates each). 

Differences highlighted on the aquagrams between salted and no salted samples are due to salt’s hydration power, mainly resulting in an increase of absorbance in the NIR range from 1400 to 1500 nm. Results suggested continuing the study of the influence of preservatives (NaCl, borax) by using the aquaphotomics approach, due to the perfect fitting of absorbance curves among replicates (Figure 2).

Figure 3 reports the aquagrams of the borax and no borax samples. 

In Figure 3, the aquagrams built up during caviar storage are reported. The absorbance of samples containing salt (NaCl) or salt + borax showed similar profiles along the storage time in the spectral range from 1340 to 1512 nm. Significant differences in absorbance values were only noted at 210 days of storage for both sets of samples, suggesting that the use of borax does not introduce significant anomalies that may result in quality deterioration during the preservation of caviar.

Applying the PCA to the whole dataset pointed out the possibility to discriminate between samples treated with salt (NaCl) and those treated with salt + borax, as shown in Figure 4.

The score plot, constructed using PC1 vs. PC2 (98.9% of explained variance), showed that, along the PC1 (83.1% of explained variance—95% of confidence level), no borax and borax samples were discriminated with positive score values associated with the borax set. In comparison, the PC2 (15.8% of explained variance) seemed to split the observations into two groups, depending on the length of storage, with lower score values associated with longer conservation time.

Figure 5 shows the primary wavelengths responsible for PCA results.

The PCA applied to the whole dataset enabled identifying borax’s presence on the 1^st^ principal component. Based on the absorbance differences in the 1400–1430 nm wavelengths range, such a result is ascribable to the first overtone bands of the OH stretching vibration of water molecules. This finding follows NaCl and borax’s different hydration capacities, resulting from their chemical structures that differ in their potential interactions with water molecules.

The LDA carried out on the calibration set (43 observations) resulted in the following linear discrimination functions: (1)Borax=−15.71 + 879.19·“1342 nm” – 724.82·“1374 nm” + 991.4·“1426 nm” 
(2)No Borax= −2.51 + 205.9·“1342 nm” – 95.23·“1374 nm” + 258.56·“1426 nm”

Overall, the classification procedure achieved a good classification rate (97.7%) (Table 2): the best classification performance regarded the observations from no borax samples, while 1 out of 20 borax samples resulted in the wrong classification.

The application of the discriminant equations (Equations (1) and (2)) to the validation set resulted in the classification matrix of Table 3:

Such high classification rates for both borax and no borax samples support the adequacy of the aquaphotomic approach in monitoring the caviar’s ripening and detecting the presence of borax.

Figure 6 shows the tenderometric data expressed (from left to right) as (i) values of the maximum peak force; (ii) the time required for the egg breaking; and (iii) the distance the plate traveled to break the eggs. 

The analysis of the two groups of samples (borax and no borax) did not point out significant differences for the consistency of the eggs (measured in terms of the maximum peak force the instrument applies to achieve eggs breakage). 

However, time and distance values to achieve the egg breakage were lower in the borax samples, suggesting a lower elasticity of the eggs receiving borax in the brining process than those treated with NaCl only. The difference in the dehydration capacity of added salts could be one reason for such different elasticity, resulting from differences in water clusters’ formation. If confirmed, this finding could affect both the shelf life duration and the end quality of the caviar. The panel test also pointed out such lower elasticity of the eggs, relating it with lower organoleptic quality of borax samples (data not shown). The preliminary results obtained by texture analyzer and sensory analysis suggests a lower quality of the caviar produced using sodium tetraborate as a preservative, supporting the need to have a rapid method to identify the sodium tetraborate presence in caviar batches.

## 4. Conclusions

The aquaphotomics approach was shown to be adequate in studying the storage process of caviar. Based on the external perturbation induced by the two preservatives on the water response, it was possible to distinguish between borax and no borax samples using a portable NIR instrument when a high S/N value is assured. The LDA applied to the validation set achieved high classification rates for both borax and no borax samples. Differences in chemical structure between the two types of salt used allowed the detection of borax, due to its different hydration power, even if added in a small percentage.

NIR spectroscopy and aquaphotomics can be, in the future, used as non-invasive methods to discriminate between fish origin, as suggested by preliminary results reported in the technical report of the project n. 201300004629, funded by Lombardy Region (data not published) and to potentially monitor the whole production chain.

## Figures and Tables

**Figure 1 sensors-20-07174-f001:**
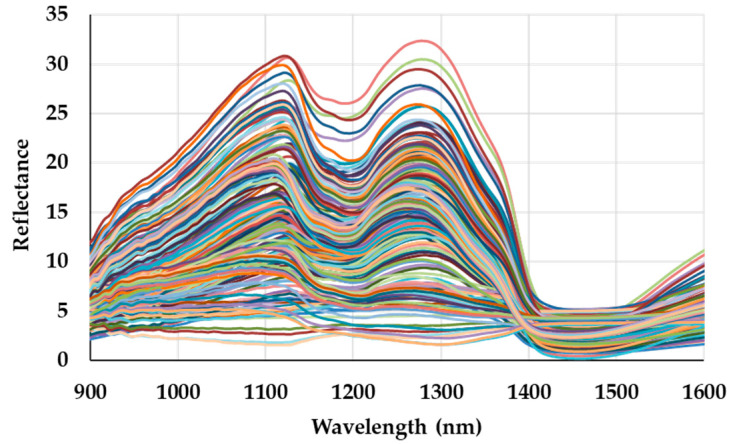
Example of caviar spectra. Each color represents a NIR acquisition (sample).

**Figure 2 sensors-20-07174-f002:**
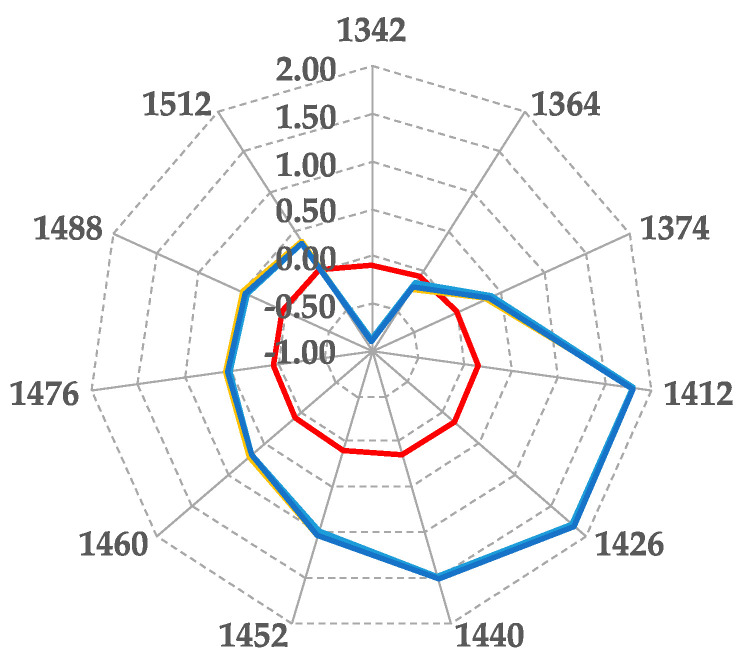
Aquagrams of samples before (red lines), and after (blue lines), salt (NaCl) addition—three replicates each.

**Figure 3 sensors-20-07174-f003:**
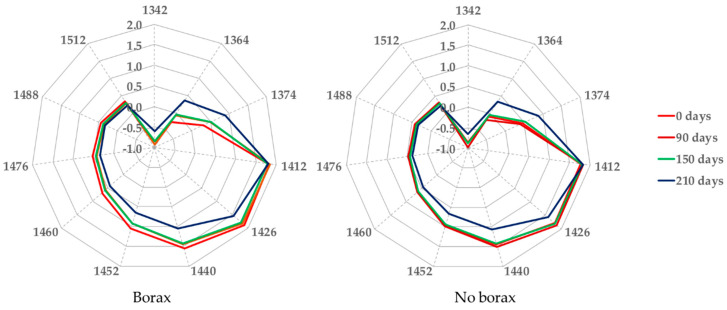
Caviar aquagrams during ripening in borax (on the left) and no borax (on the right) samples. The red lines (bright and dark) refer to acquisitions at 0 and 90 days; the green lines represent samples at 150 days, and the blue lines the samples at 210 days from packaging.

**Figure 4 sensors-20-07174-f004:**
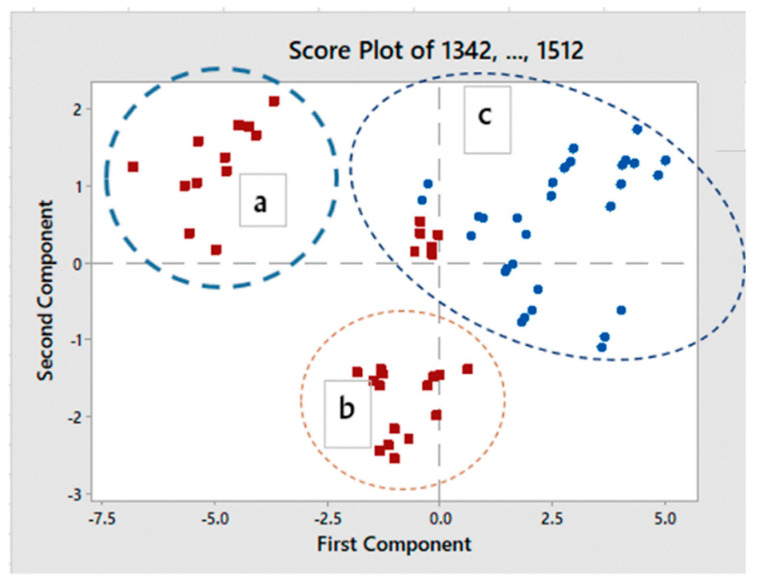
Principal component analysis (PCA) applied to the whole caviar sample set in the aquagram range: NaCl salt (square), borax (circle), a = short time of storage; b = long time of storage; c = whole borax set.

**Figure 5 sensors-20-07174-f005:**
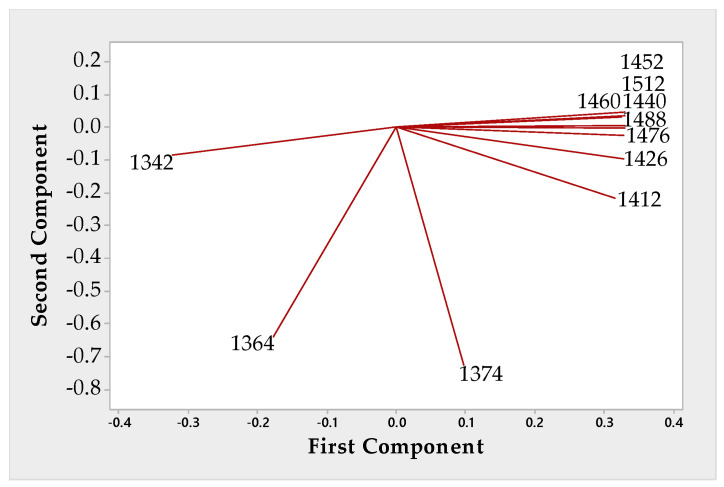
PCA loading plot.

**Figure 6 sensors-20-07174-f006:**
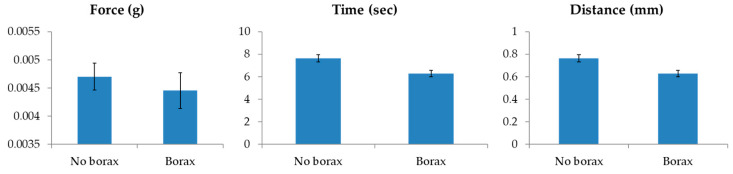
Results of the tenderometric analysis.

**Table 1 sensors-20-07174-t001:** Concentration ranges of the main constituents of the matrix [12].

Constituent (%)	Min	Max	Average	SD
Moisture	52.10	58.53	55.46	1.43
Protein	22.48	27.06	24.75	1.38
Fat	13.64	18.46	16.41	1.32
Ash	1.31	4.20	3.47	0.71

**Table 2 sensors-20-07174-t002:** Classification matrix of the calibration set.

	Borax	No Borax
**Borax**	19	0
**No borax**	1	23
**Total**	20	23
**Correctly classified**	19	23

**Table 3 sensors-20-07174-t003:** Classification matrix of the validation set.

	Borax	No Borax
**Borax**	10	0
**No borax**	0	12
**Total**	10	12
**Correctly classified**	10	12

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
