# Peer review of "The Influence of the Presence of Borax and NaCl on Water Absorption Pattern during Sturgeon Caviar (Acipenser transmontanus) Storage"

_sensors, 2020, doi:10.3390/s20247174_

Round 1
Reviewer 1 Report
The manuscript "The influence of the presence of borax and NaCl on water absorption pattern during caviar (Acipenser transmontanus) storage" is in scope of "Sensors", and could be potentially of interest to its readers, however, it should undergo a major revision.
1. It is not clear how this tool could be used for detection of borax use. There is no classification accuracy of samples outside the training set given for PCA (in fact, it seems that only the grouping has been done, as there is nothing abot the division of samples into training and and test sets), and the
aquagrams are very similar for all samples (what is stated by the authors as well). So, other discrimination and/or regression models should be checked (e.g. SIMCA or PLS-DA), with the correct classification rate, and false positives/negatives concerning the detection of borax addition.
2. As the portable instrument has been used, and portable instruments usually have a relatively low resolution, the question is whether the resolution was high enough to separate closely located water absorbance bands. Usually the resolution of 0.5 or 1 nm is advised. So, this should be adressed by the
authors.
3. In Fig. 3 a better differentiation of lines corresponding to various days should be done - these grays are very similar, and as the lines are overlapping, this figure is hard to analyze. Maybe some dotted lines
would be easier to discern.
4. In Materials and Methods there is nothing abot the analysis of matrix composition, which is reported in Table 1. So, where the compositional data comes from?
5. A discussion concerning the relation (if there is any) between aquaphotomics and tenderometric data should be added to the manuscript, as in this form they are desribed separately and it is not clear why
these methods have been combined in one work.
Reviewer 2 Report
This is a paper that used a spectroscopic technique in order to gain a publication in Sensors or another spectroscopic journal, but there is no valid reason given for the research. No good argument is given as to why the Italian industry requires a spectroscopic means of detecting borax, and how this actually helps development of a borax-free product. The only reason to use a spectroscopic technique is to determine if borax has been added to a sample of unknown or questionable origin. If a sample in Lombard facilities has borax added and this is known, then it will be marked. This type of spectroscopic evaluation is only necessary and useful in order to monitor product safety and quality in sample groups from various origins; such as imports. It is not even clear whether borax should or should not be in the product. No background for caviar processing and how the ripening process is accomplished is given.
Caviar should be identified as sturgeon caviar in the title, abstract and introduction as well as the Latin binomial.
Percentages are to be given with two significant figures only.
NIR is the typical abbreviation, not NIRS
Parenthetical expressions are over-utilized in the Introduction and not needed. (about 12-15 years), (Italy) and (the Lombard production capacity of caviar exceeds 40 tons of which about 90% is destined for export for a value of about 15,000,000 euros) is important information that belongs in the text as is, not in ().
Line 48-49 should read offer an “endproduct”
Line 52 delete “the” before Italian caviar
Line 55 states that some boron compounds are still allowed but only borax is mentioned or discussed. If there are others used, what are they?
Line 58 should read international; no capitalization
Lines 62 to 68 are unclear and need rewriting. This type of spectroscopic evaluation is only necessary and useful in order to monitor product safety and quality in sample groups from various origins; such as imports.
Line 73 be consistent with Euro or US enumeration; use . or , not both. (3.5% NaCl) or salt + borax (3,5 % NaCl + 0.04‰ E285) Also what is E285? Your readers are not going to know this!
Figure 1 is useless. There is no legend or indication of why the spectra are different.
Fig 6 is not needed: readers know what salt and water are.
Fig 7 is hard to read and has no legend. What significant and relevant data are reported here?
Line 174 and 175 use the word “thesis” which makes no sense.
The final statement Line 183 FF is absurd in its far-reaching claims; nothing in this work can possibly be used to discriminate fish origins.
“NIR spectroscopy, and Aquaphotomics, can be in a next future used as noninvasive method also todiscriminate between fish origin and to monitor the whole production chain.”
Round 2
Reviewer 1 Report
In my opinion, the manuscript was improved enough to warrant a publication in Sensors as a letter. It shows an interesting and plausible perspective of detecting an undeclared/unwanted borax detection with reasonable credibility.
Author Response
Thanks for your positive evaluation.
According to your suggestion, English has been revised, and the revisions are reported in clear.
Reviewer 2 Report
Much improved justification for the research was added. The paper still requires some English editing but can be published after that
Author Response

(The authors gave the same response as above.)
